# Saliency Suppressed, Semantics Surfaced: Visual Transformations in Neural Networks and the Brain

**Gustaw Opiełka**[*]  **Jessica Loke**   **Steven Scholte**
University of Amsterdam

## Abstract

Deep learning algorithms lack human-interpretable accounts of how they transform raw visual input into a robust semantic understanding, which impedes comparisons between different architectures, training objectives, and the human brain. In this work, we take inspiration from neuroscience and employ representational approaches to shed light on how neural networks encode information at low (visual saliency) and high (semantic similarity) levels of abstraction. Moreover, we introduce a custom image dataset where we systematically manipulate salient and semantic information. We find that ResNets are more sensitive to saliency information than ViTs, when trained with object classification objectives. We uncover that networks suppress saliency in early layers, a process enhanced by natural language supervision (CLIP) in ResNets. CLIP also enhances semantic encoding in both architectures. Finally, we show that semantic encoding is a key factor in aligning AI with human visual perception, while saliency suppression is a non-brain-like strategy.

## 1 Introduction

Natural and artificial visual systems rely on transformations of raw visual inputs (retinal or pixels) into abstract representations. Both human (DiCarlo & Cox, 2007; DiCarlo et al., 2012) and computer vision algorithms (Zeiler & Fergus, 2013) learn increasingly abstract features along the visual information processing chain - from low-level (e.g., oriented edges) through mid-level (e.g., simple shapes), high-level (e.g., object categories) to, ultimately, rich semantic properties (Doerig et al., 2022). Successful visual systems should perform these transformations in a way that is robust to variations in viewing conditions (Hinton, 1987). For instance, humans will not have a problem recognizing a street sign even though it's snowing. However, neural networks' robustness against adversarial attacks and real-world noise has traditionally lagged behind human vision (Goodfellow et al., 2015; Hendrycks & Dietterich, 2019). Despite achieving impressive results on IID (independent and identically distributed) test data, their performance dips significantly on out-of-distribution (OOD) benchmarks (Geirhos et al., 2020), showing that the visual transformations in artificial visual systems still have a long way to go.

Recent advances with networks having different architectural biases and novel training objectives have been closing this robustness gap (Geirhos et al., 2021). CLIP, which integrates both an image and text encoder (Radford et al., 2021), is at the forefront of this progress. CLIP distinguishes itself by its unique training approach: rather than training the image encoder solely on object categories, it is designed to align image representations with their corresponding textual descriptions. This is vital because image descriptions provide more than just object enumerations; they convey deeper semantic insights, spatial relations, and the context of scenes (Doerig et al., 2022). CLIP's training methodology resulted in notable improvements across multiple downstream metrics, including enhanced robustness against distortions, a notable departure from the outcomes typically seen in standard object classification networks (Radford et al., 2021; Geirhos et al., 2021). Moreover, CLIP, and other language-aligned models, also build better models of the visual cortex (Wang et al., 2022)

---

[*]Correspondence to g.j.opielka@uva.nl. Open-source code and data available at `https://github.com/gucioopielka/Saliency-Semantic-RSA`

and exhibit error patterns more aligned with those of humans (Geirhos et al., 2021). Jointly, these findings hint at a computational difference in how language-aligned networks process visual data. Specifically, increasing access to semantic information during training, appears to help networks learn visual features that encapsulate high-level image details while being resilient to distortion.

While this suggests that there are architectural and training objective differences in the transformation of visual information in neural networks, how can we study them? Traditional methods, such as visualizing individual neuron features or creating attention maps (Olah et al., 2020), offer limited insights. They rely on qualitative assessments of human-interpretable concepts and fail to explore intermediate distributed representations, highlighting the need for techniques that provide quantitative metrics and access to collective neuronal activities. To address these limitations, we draw inspiration from neuroscience, specifically representational similarity analysis (RSA; Kriegeskorte, 2008), a technique employed to correlate model representations with neural data. This approach can be used to quantify the alignment between image attribute representations with neural network activations, offering a novel post-hoc interpretability tool. By crafting pairwise similarity matrices, RSA quantifies the extent of stimulus information decodable from network activations, thereby providing insight into the distributed encoding of specific attributes.

In this study, we examine the visual transformations in neural networks and the human visual cortex by considering two image properties residing at opposite ends of the transformation hierarchy - visual saliency and semantic similarity. Both provide informative descriptors of visual scenes and represent, respectively, low-level, and high-level image properties with minimal overlap. Moreover, they have a recognized significance for visual information processing in humans. Visually salient items are known to be behaviorally distracting for human observers (Theeuwes, 2010) and have an established neural basis (Treue, 2003), while recent studies have shown that semantic similarity has been shown to explain well both behavioral judgments (Marjieh et al., 2022) and activity in both early and late visual cortex (Doerig et al., 2022).

Our paper is divided into three main sections. First, we study how neural networks represent salient and semantic information. Furthermore, we introduce a custom dataset where we systematically manipulate salient and semantic information in images. This analysis yields new insights and uncovers architectural and training objective differences in how neural networks perform visual transformations. We then turn to the human brain, where we compare where biological visual transformations coincide and where they differ with those in artificial systems. Finally, we use our newly gained knowledge of saliency and semantic representations in neural networks to examine what does and doesn't drive the alignment between network and brain representations.

## 2 SALIENCY AND SEMANTICS IN NEURAL NETWORKS

### 2.1 REPRESENTATIONAL ALIGNMENT BETWEEN NETWORK AND IMAGE FEATURES

#### 2.1.1 QUANTIFYING LOW- AND HIGH-LEVEL IMAGE PROPERTIES

We operationalize low-level image features as visual saliency maps, as described by Itti et al. (1998). Saliency at a given location is defined by how different this location is from its surround in color, orientation, and intensity. This results in saliency maps - purely visually informative regions, not influenced by high-level features such as objects, or relation between objects (i.e. saliency maps should not have any semantic component). While there are many low-level image descriptors available, we choose visual saliency for its ability to capture the conjunction of multiple early visual features, as well as for their recognized importance in human visual processing.

To describe semantic information in images, we make use of image captions embeddings. We take advantage of the fact that COCO dataset offers five distinct captions per image, each provided by a different human annotator (e.g., "A dog is playing with a toy"). Using the Universal Sentence Encoder (USE; Cer et al., 2018), we transformed these captions into 512-dimensional vectors and took the average of all captions for each image. Taken together, we have quantifiable measures of image content at a low-level of abstraction (*Visual Saliency*) and high-level of abstraction (*Semantic Similarity*).

a. Computing Baseline Saliency/Semantic RSA

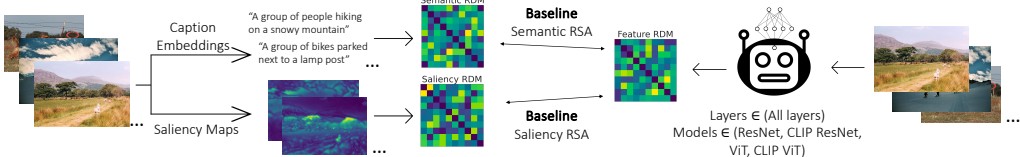

b. Computing Saliency/Semantic ΔRSA

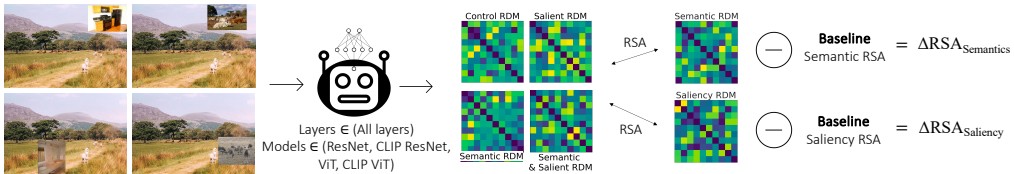

Figure 1: Experimental approach. (a.) Calculating the alignment between network features and visual saliency/semantics. Saliency maps and caption embeddings from the COCO images were converted into RDMs. Their correlation with network feature RDMs establishes the degree of alignment - Saliency/Semantics RSA. (b.) Sensitivity to distractors. Network features were extracted from images with all 4 distractor types (see Figure 3). These RDMs were correlated with the original saliency and semantic RDMs (from a) to establish RSA resulting from seeing the distractors. Taking the absolute difference with the baseline RSA (from a) we get ΔRSA Saliency/Semantics which measures network alignment to original low- and high-level image content amidst distractors.

### 2.1.2 RESULTS

We assess the alignment between neural representations and saliency/semantics by first converting activations, saliency maps, and caption embeddings into representational dissimilarity matrices (RDMs). We then quantify the degree of alignment using non-parametric regression. See 1a. for visualisation, and Appendix A.1 for further details and formalization of this method. Figure 2 showcases the representational alignment of features extracted from ResNets and ViTs trained on object classification tasks (ImageNet; Deng et al., 2009) and CLIP objectives. These features were derived from a dataset of 1150 images from the COCO database, along with their corresponding saliency maps and caption embeddings.

We find both training objective and architectural differences in the representation of saliency and semantics in neural networks. First, CLIP training enhances the semantic information in both architectures, giving support for the intuition of previous work on the effects of natural language supervision (Ghiasi et al., 2022; Wang et al., 2022). There is also difference in the trend of how semantic information progresses across the network depth between ResNets and ViTs. While in ResNets, the semantic information seems to grow more or less linearly, some ViTs exhibit a U-shaped pattern, with the semantic information peaking in the late-mid layers and then dropping off in the latest layers. Since smaller ViT models, B16 and B32, showed similar patterns, we only show the latter for easier visibility. See 9 to see all tested ViTs.

Regarding saliency, CLIP seems to have an effect primarily in ResNets. Specifically, in the earliest layers, CLIP-trained ResNets show negative alignment with saliency maps, lower than those of ImageNet-trained ResNets. We hypothesize that these values represent *suppression* of salient information in the earliest layers. We note, that when the values are averaged across many layers, the saliency RSA values for ImageNet's ResNet hover around 0, but raw data shows that there are still layers with stronger negative saliency suppression. In CLIP's ResNet, however, there were more of these layers. This suggests that suppression of saliency might be a common strategy employed by ResNets, which is enhanced by natural language supervision.

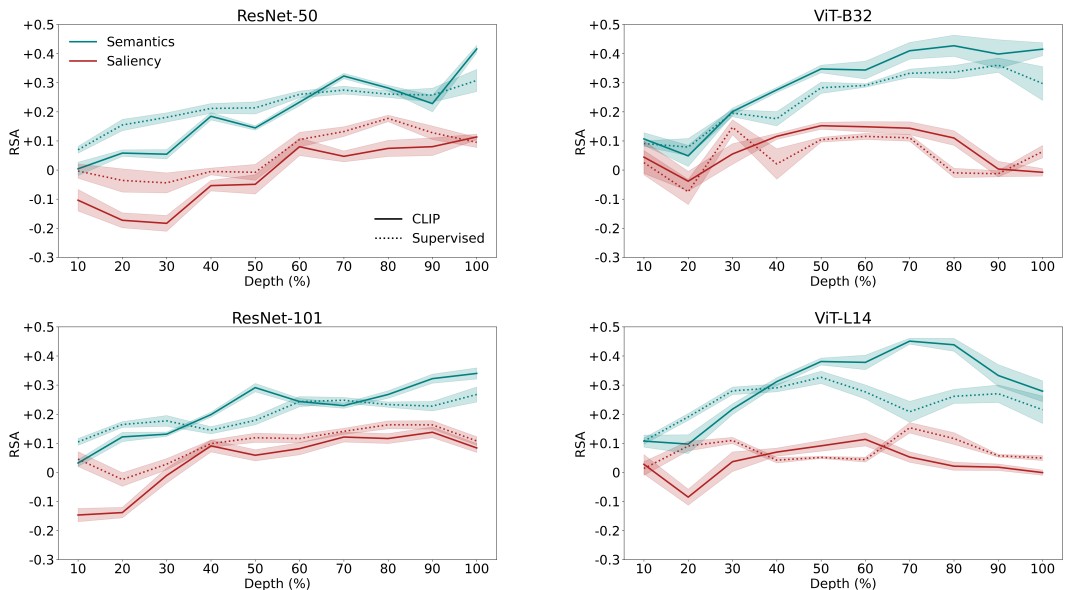

Figure 2: Representation of semantic information generally increases as a function of layer depth in all networks. CLIP enhances the amount of semantic information represented by the networks, mostly in later layers. Notably, however, we see a negative alignment between layer representations and saliency maps in ResNets trained with CLIP, suggesting saliency suppression. Architecturally, ResNets encode more saliency information than ViTs ($p < .001$). Note: for visualization purposes, the values are averaged across several layers. To view the raw values, see Appendix A.3

## 2.2 CAUSAL EFFECTS OF SALIENCY AND SEMANTICS

Our exploration so far provides correlational evidence for differences in the encoding of saliency and semantic information. To investigate how intermediate layers directly react to salient and semantic information, we systematically manipulated saliency and semantic similarity in the images. To this end, we created a custom dataset.

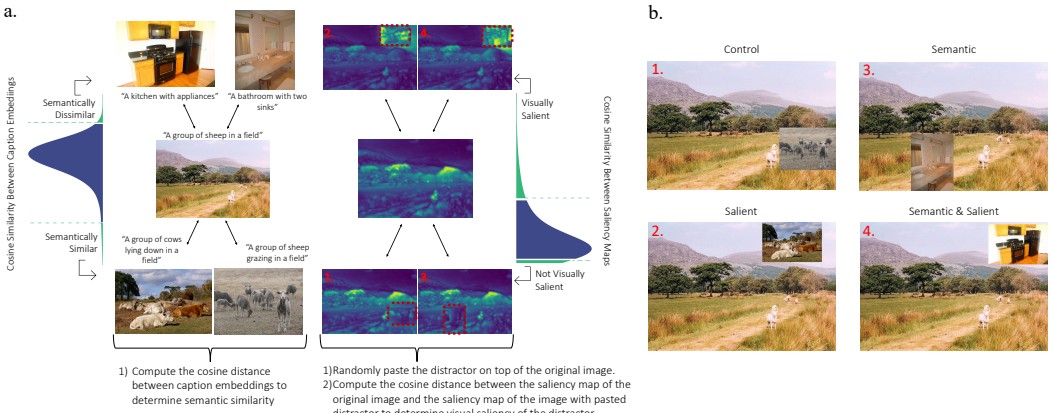

Figure 3: Visual and semantic distractors. (a.) Left: For the central image (target), the top images have dissimilar captions and the bottom ones are similar. Right: Saliency maps of the target image and images with distractors. Top images alter the target's salient features more than the bottom ones. Red outlines indicate distractor locations. Distributions on the left and right illustrate semantic similarity and visual saliency thresholds, respectively. (b.) Images with four distractor types, with numbers corresponding to saliency maps in a.

### 2.2.1 CREATING THE DATASET

We formulated a technique to introduce distractors by overlaying a smaller image onto a larger one. This smaller image is resized to occupy 10% of the larger image's area while preserving the original aspect ratio of both images. The positioning of the smaller image (distractor) is randomized yet always bordering the edge of the larger image (target). This deliberate placement at the periphery is intended to minimize the likelihood of the distractor obscuring the most informative regions of the target image. Below, we delineate criteria to categorize whether the distractor significantly changes the distribution of saliency or semantic information in the target image.

**Saliency** We consider the distractor to be visually distracting if it causes big changes to the distribution of visual saliency in the target images. To quantify this intuition, we generated saliency maps for both the unaltered images and the images containing the distractor and compute cosine distance between them. Large cosine distance indicates that the distribution of saliency in the altered image significantly differs from the unaltered, target image. After repeating this procedure for a sample of 1,000 image pairs, we derive the distribution of saliency similarity. We choose the 5th and 95th percentiles of this distribution to define our similarity thresholds. Accordingly, distractors falling below the 5th percentile demonstrate minimal visual distraction, while those above the 95th percentile are deemed visually distracting.

Figure 3 provides insight into our approach. In the images with a non-salient distractor (images 1 and 3), the distractor blends with the background, making its low-level features merge with the main image. This means the altered image's saliency map is almost identical to the original. However, in images with a salient distractor (images 2 and 4), the distractor stands out and significantly changes the distribution of salient features in the image.

**Semantics** To estimate the distribution of semantic similarity of the images in the COCO dataset, we performed pairwise comparisons of 5000 image caption embeddings, again using cosine distance. For this distribution, we took the 1st and 99th percentiles to define our similarity thresholds. Consequently, an image from the COCO dataset (target) whose caption similarity to the target image ranks below the 1st percentile is seen as semantically similar, whereas one above the 99th percentile is highly dissimilar, thus qualifies as a *semantic distractor*. Taken together, this allows for a controlled experimental dataset with 4 different types of distractors:

**Control** Is not visually salient and semantically similar to the target. Essentially allows for testing the effect of pasting an image on top of a target image.

**Salient** Is visually salient and semantically similar to the target.

**Semantic** Is not visually salient, but is semantically dissimilar to the target.

**Salient & Semantic** Is visually salient and semantically dissimilar to the target. Allows for investigating the effects of the interaction of semantics and saliency.

We created a dataset of 1150 images, each having every type of distractor (in total $1150 \times 5$ images, including baseline images without distractors).

### 2.2.2 RESULTS

We measure what effect a distractor had on the network representations of saliency and semantics by computing network activations RDMs when presented with a distractor, and then correlating it (Spearman's Rho, $\rho$) with saliency/semantics RDMs of the *original* images. This approach is based on the premise that if a distractor doesn't influence the network, its representation of the image should remain consistent, whether the distractor is present or not. Formally, given RDMs of network activations ($\mathbf{D}_a$), saliency maps ($\mathbf{D_m}$), and image captions ($\mathbf{D_c}$), we calculate baseline RSA:

$$\text{Base RSA}_{\{m,c\}} = \rho(\mathbf{D}_a, \mathbf{D}_{\{m,c\}})$$

And RSA with distractors (Dist RSA) for $d \in \{$Control, Salient, Semantic, Semantic & Salient$\}$:

$$\text{Dist RSA}_{\{m,c\},d} = \rho(\mathbf{D}_{a,d}, \mathbf{D}_{\{m,c\}})$$

Note that $\mathbf{D}_m$ and $\mathbf{D}_c$ remain constant across $d$. The $\Delta$RSA is then the absolute difference:

$$\Delta\text{RSA}_{\{m,c\},d} = |\text{Dist RSA}_{\{m,c\},d} - \text{Base RSA}_{\{m,c\}}|$$

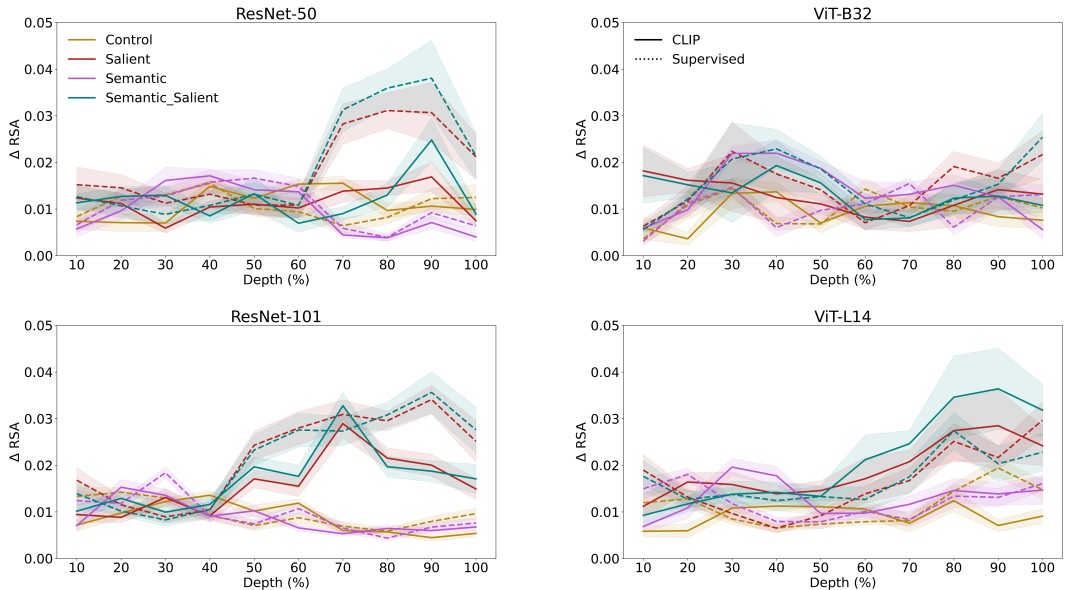

Figure 4: $\Delta$RSA Saliency. Higher $\Delta$RSA values values indicate greater disruption in network saliency representations as a result of 4 different distractors (Control, Salient, Semantic, and Semantic & Salient). We see an effect of training objective in ResNets: salient distractors cause more disruption in later layers of ImageNet-trained ResNets compared to those trained with CLIP. We also see an architectural difference: smaller ViTs (see Figure 10 to view ViT-B16), show less disruption than ResNets to salient distractors. In the largest CLIP-trained ViT there is an interaction effect of saliency & semantics.

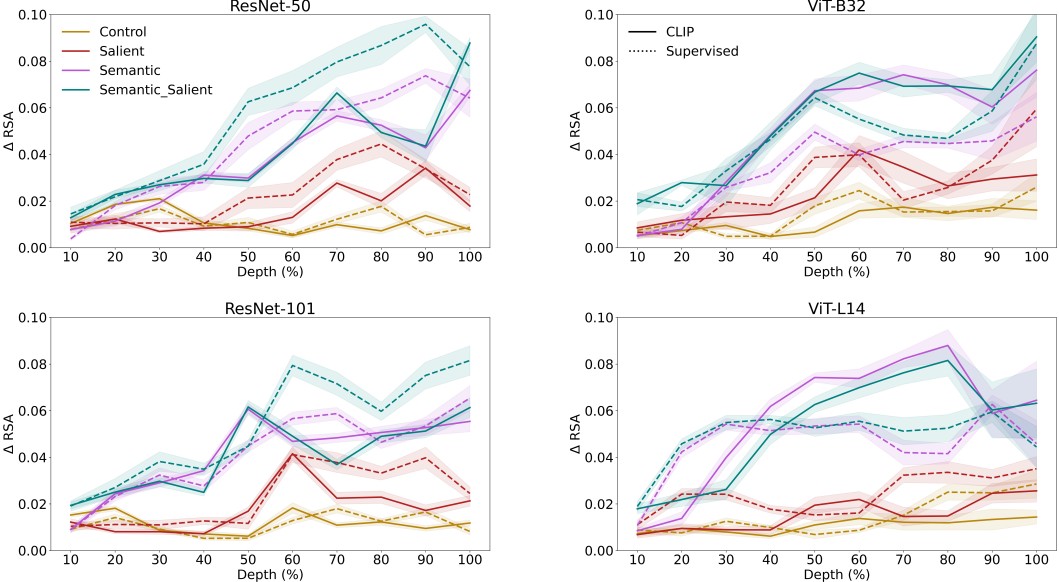

Figure 5: $\Delta$RSA Semantics. Higher $\Delta$RSA values signal increased disturbance in network semantic representations. The effect of training objectives varies per architecture. While, ImageNet-trained ResNets in general both salient and combined distractors significantly disrupt semantic representations more than in ViTs, a trend mitigated by CLIP training. Architecturally, CLIP training reduces disturbance in ResNets while increasing it in ViTs.

**Saliency** In Figure 4, we illustrate the impact of distractors on the network's saliency representation, highlighting key differences based on training objectives and architecture. Specifically, for ResNets, salient distractors significantly disrupt saliency representation in models trained for object classification compared to those with CLIP. The similar patterns observed for both purely salient and mixed salient-semantic distractors indicate that the disturbance is caused solely by saliency. This effect is particularly pronounced in the network's later layers, supporting our interpretation of negative RSA values in early layers of ResNets as saliency suppression, enhanced by CLIP (discussed in Section 2.1.2 and illustrated in Figure 2). We propose that this enhanced suppression leads to a more pronounced constraint on the representation of visual features in subsequent layers, which may, in turn, influence the network's ability to generalize from visual input to higher-level concepts.

For ViTs, the response to salient distractors is markedly different; both ImageNet and CLIP networks exhibit resilience to salient disruptions. An exception is observed in the largest model, ViT-L14, where CLIP training appears to heighten sensitivity to a mix of salient and semantic distractors, more than in its ImageNet-trained counterpart. This observation likely stems from CLIP's impact on semantic representation in ViTs—a topic we explore further in the subsequent paragraph—rather than saliency, which remains consistent. Coupled with earlier findings that ViTs encode less saliency information (refer to Section 2.1.2 and Figure 2), this pattern hints at an inherent architectural trait of transformers being less sensitive to saliency compared to CNNs.

**Semantics** In Figure 5, we visualize the effect distractors had on semantic representation. Similar to saliency, we see both architectural and training objective differences. First, we note that in Image-Net trained networks, the combination of semantic and salient distractors, and to a lesser extent purely salient distractors, causes greater disturbance in ResNets than in ViTs. Specifically, in ResNets there appears to be an additive effect of saliency on top of the disturbance caused by the semantic distractors. This effect disappears with CLIP training, where the semantic and salient distractors cause the same disturbance as the purely semantic distractors, meaning that the disturbance in the representations is driven purely by semantic dissimilarity.

Interestingly, the additive effect of saliency on top of semantic dissimilarity is not present in ViTs trained with both object classification and CLIP objectives. Instead, CLIP seems to sensitize ViTs semantic representation to semantic dissimilarity, with semantic distractor causing a greater disturbance in CLIP-trained models, than in those trained with ImageNet. This enhanced semantic sensitivity might have explained the outlier finding in ViT-L where the saliency representation was more disturbed by semantic and salient distractors. Together, this again strengthens our conclusions on ResNets being more sensitive to saliency than ViTs and natural language supervision suppressing this sensitivity. Finally, it also shows how low-level visual processing can disturb high-level processing (and vice-versa), and how natural language supervision can modulate this effect.

**Performance** Effects of the distractors on network classification performance mostly mirror the representational results (see Appendix A.4; Table 1). Semantic distractors cause more disturbance in networks' performance than salient ones. CLIP-trained models suffer more from semantic distractors than Image-Net-trained networks. However, we note that our method might underestimate the effects of salient distractors on network performance. As shown in our Grad-Cam exploration (Figure 11), networks often attend to the salient distractors, but since the semantic component is kept similar to the target image, they still make the correct classification.

## 3 SALIENCY AND SEMANTICS IN THE BRAIN

We used the brain responses from open-source Natural Scenes Dataset (NSD; Allen et al., 2022) of human subjects viewing COCO images (see Appendix A.2 for details). For our analysis, we used the subset of 872 images viewed by all participants. We performed RSA between representational spaces of neural responses from different ROIs along the visual hierarchy, and saliency/ semantics, similarly to neural networks (section 2.1.2). We call the RSA values Brain Scores.

### 3.1 RESULTS

In Figure 6, we show how the brain represents saliency and semantics along the cortical hierarchy, ranging, roughly, from low-level visual cortex, such as V1-V3, to higher-order areas, such as parahippocampal place area (PPA).

Figure 6: Saliency and semantic representation in the brain. Similar to neural networks (as shown in Figure 2), the deeper into the visual processing hierarchy, the more the neural representations align with semantics. In contrast, saliency exhibits a different pattern: it is most pronounced in the early cortex and diminishes in higher processing stages. Notably, unlike in neural networks, we do not observe a negative alignment with saliency.

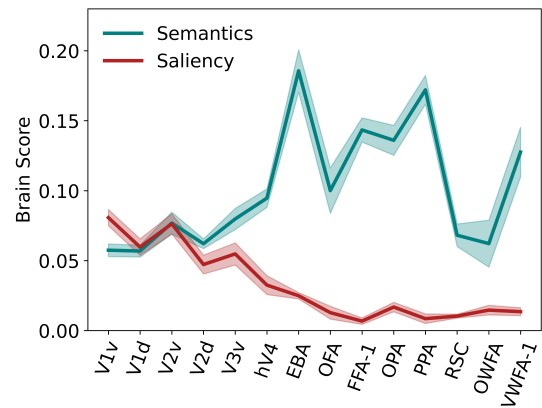

## 4 ALIGNMENT BETWEEN NEURAL NETWORKS AND THE BRAIN

In this section, we leverage our visual saliency and semantic similarity metrics to investigate how visual transformations shape particular layers' alignment with neural data.

### 4.1 RESULTS

Using the shared images in NSD, we computed the representational alignment between brain data and network features extracted from all networks and all their intermediate layers. We then correlated the Semantics and Saliency RSA scores of the layers with their Brain Scores. We observed a high correlation coefficient of 0.83 ($p <$.001) for Semantic RSA and Brain Scores. This suggests that representational alignment between brain and network-extracted features is to a large extent driven by the amount of semantic information encoded in the network layers.

The correlation coefficient between Saliency RSA and Brain Scores was 0.64 ($p <$.001). While this would suggest that encoding of saliency is also predictive of the Brain Scores, as we can see from the plot on the right in Figure 7 the relationship is not linear for all values of Saliency RSA. If we split the data into negative and positive values of Saliency RSA, we see that the trend is only linear for negative values ($r = 0.67$, $p <$.001, $n = 414$) and much weaker for positive values ($r = 0.11$, $p = $.019, $n = 1087$). This suggests that the larger the suppression of salient features in a specific layer, the less aligned it is with brain representations.

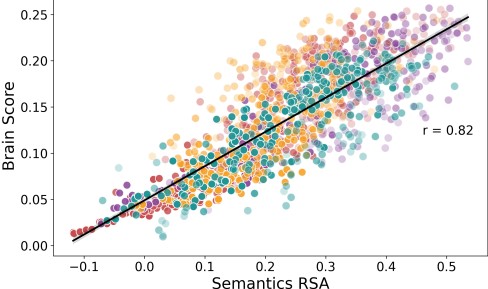 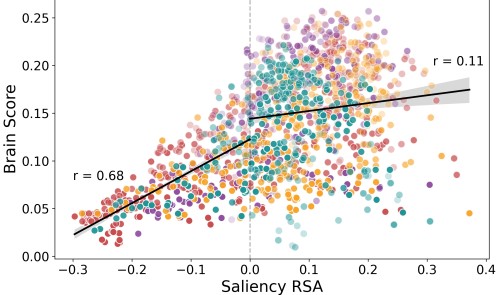

Figure 7: Scatter plots of Brain Scores as a function of alignment with semantics (left) and saliency (right), for all layers (n = 1505), and networks (ImageNet ResNet, CLIP ResNet, ImageNet ViT, and CLIP ViT). The degree of layer's alignment with semantics is strongly predictive of its alignment with the visual cortex (Left). Layers that show negative alignment with saliency, negatively predict brain activity (Right). Brain Score denotes layer's alignment with all vertices in the occipitotemporal cortex (OTC).

## 5 DISCUSSION

**Motivation**  Failure of classic neural networks on out-of-distribution images compared to human performance and novel training objectives (such as CLIP; Geirhos et al., 2021) can be conceptualized by differing transformation of low-level image features to high-level semantic understanding. By using RSA as an interpretability tool, we have quantitative, yet human-perceptible metrics to examine what strategies different visual systems use. Moreover, through a controlled dataset, we can discern the causal relationships of how low-level processing affects high-level processing and vice versa.

**Key Findings**  (1) There are architectural biases linked to processing, both saliency and semantics. In particular, CNNs seem to be more affected by saliency than Visual Transformers, while both architectures encode semantics to a large extent. (2) On a representational level, ResNets deal with salient information by suppressing it in early layers. (3) Natural language supervision (CLIP) seems to enhance saliency suppression and enhance semantic encoding in both CNNs and Transformers. (4) The way neural networks process semantics shows similarities with the human visual cortex, suggesting a convergence in representations at a high-level of abstraction. However, the strategy of suppressing salient information appears to be unique to neural networks and not observed in human visual processing.

**CLIP Limitations**  Our analysis of natural language supervision's impact via CLIP is limited by its training on a significantly larger and more diverse dataset (around 400M images) compared to ImageNet (1.5M images). This concern is accentuated given that dataset diversity has been suggested to be a key factor in aligning AI with human visual perception (Conwell et al., 2022). Our control analysis (Figure 12 and 13) shows that expanding dataset size alone, as seen in a ResNet-50 trained on 940M images with Semi-Weakly Supervised Learning (SWSL; Yalniz et al. (2019)), does not fully replicate CLIP's effects on saliency and semantics. Nevertheless, the nature of SWSL's web-scraped images, potentially more complex than ImageNet's, introduces uncertainty about our findings. CLIP's effects may derive more from its exposure to complex visual scenes, rather than its integration with language.

**Machine learning insights**  We find that CNNs appear more sensitive to saliency than ViTs. This distinction can be attributed to the inductive biases of each architecture: ResNets excel in processing local image features, whereas ViTs are better equipped to capture long-range dependencies, with the latter having larger receptive fields than the former (Raghu et al., 2022; Khan et al., 2023). Given that visual saliency largely involves local contrast detection (Itti et al., 1998), it is plausible that ResNets are inherently biased towards salient visual features. This might have caused ResNets to suppress saliency to a larger extent than ViTs in order to accommodate semantic information.

**Neuroscientific Insights**  Our study suggests the alignment between neural and model representations is primarily influenced by the depth of semantic encoding, aligning with theories that the visual cortex's main function is to transform raw visual input into semantic representations (Doerig et al., 2022; Bracci & Op De Beeck, 2023).

Our findings about CLIP suppressing saliency information to obtain more robust visual representations give additional insights into the work on comparing error patterns in humans and machines. Specifically, Geirhos et al. (2021) found that ResNet-50 trained with CLIP makes non-human errors, even though it is more robust to OOD images than its Image-Net trained counterpart. Potentially, this could be caused by the non-brain-like enhancement of saliency suppression.

Furthermore, the observed low alignment between saliency and cortical activity, at times even below semantic alignment in early visual regions, suggests potential limitations of fMRI in capturing early visual processes, critical for saliency (Van Humbeeck et al., 2018). This highlights the need for future studies using higher temporal resolution techniques, like EEG, to probe the representational saliency-cortex alignment more accurately.

**Future Directions**  The discovery of saliency suppression stands out as an unexpected and intriguing finding. To deepen our understanding, we propose to extend our research through (1) Circuit-level analyses (2) Establishing formal links between saliency maps and CNN computations (3) Assessing neural network responses to a variety of low-to-mid-level image descriptors, beyond saliency, to achieve a comprehensive understanding of visual transformations.

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

# A  APPENDIX

## A.1  REPRESENTATIONAL SIMILARITY ANALYSIS (RSA)

We analyzed neural network activations ($a_s$), vectorized saliency maps ($m_s$, $256 \times 256$ sized), and 512-dimensional caption embeddings ($c_s$) for each image $i$. Pairwise dissimilarities between images $i, j$ were calculated using cosine distance for activations ($d_a$), saliency maps ($d_m$), and captions ($d_c$):

$$d_{\{m,a,c\}}(i,j) = 1 - \frac{\boldsymbol{x}_s \cdot \boldsymbol{x}_t}{\|\boldsymbol{x}_s\|_2 \|\boldsymbol{x}_t\|_2}, \text{ where } \boldsymbol{x} \in \{\boldsymbol{m}, \boldsymbol{a}, \boldsymbol{c}\}$$

We constructed three representational dissimilarity matrices (RDMs) ($\mathbf{D}_a$, $\mathbf{D}_m$, $\mathbf{D}_c$) to represent activations, saliency, and captions.

$$\mathbf{D} = \begin{bmatrix} d(1,1) & d(1,2) & \dots & d(1,n) \\ d(2,1) & d(2,2) & \dots & d(2,n) \\ \vdots & \vdots & \ddots & \vdots \\ d(n,1) & d(n,2) & \dots & d(n,n) \end{bmatrix}$$

Given the symmetric nature of RDMs, the analysis focused on the upper triangle matrices. Spearman rank correlation coefficients ($\rho$) assessed relationships among RDMs, evaluating the alignment between neural activations and the representational spaces of saliency maps ($\text{RSA}_m$) and image caption embeddings ($\text{RSA}_c$).

## A.2  NSD DATASET

We employed the Natural Scenes Dataset (NSD; Allen et al., 2022), a publicly available collection of high-resolution 7T fMRI responses to approximately 10,000 distinct natural scene images from eight participants. These images, derived from the COCO dataset, were square cropped and displayed for 3 seconds each at a visual angle of 8.4° x 8.4°, separated by 1-second intervals. Participants were asked to focus on a central fixation point and to press a button to indicate recognition of any repeated image. Further details on the NSD are available at its website: `http://naturalscenesdataset.org`. Our study focused on the Algonauts Challenge subset of the NSD dataset, which notably excludes subject 7, differentiating it from the complete NSD collection (Gifford et al., 2023). This analysis utilized a specific set of 872 images that were viewed by all subjects, along with the neural response data from the entire occipitotemporal cortex (OTC).

### A.3 NETWORK SALIENCY/SEMANTIC RSA

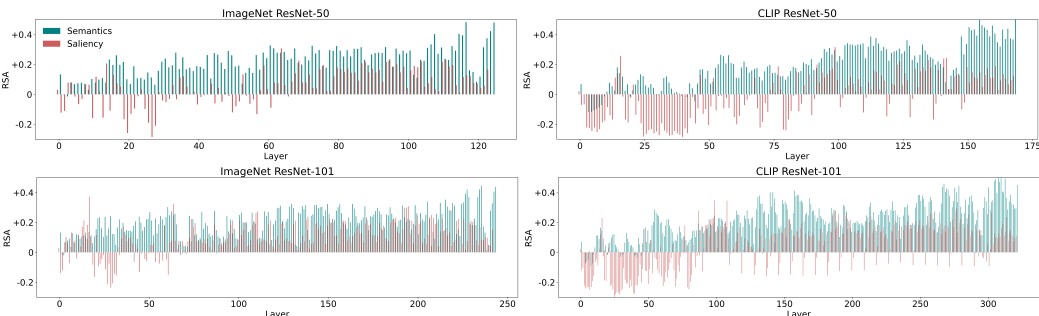

Figure 8: RSA values for all layers in ResNets trained with ImageNet (left) and CLIP (right).

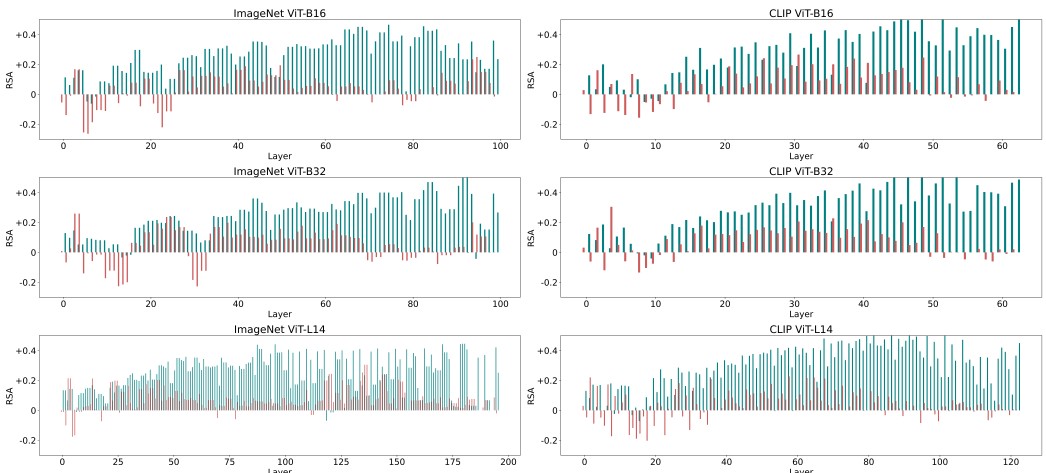

Figure 9: RSA values for all layers in ViTs trained with ImageNet (left) and CLIP (right).

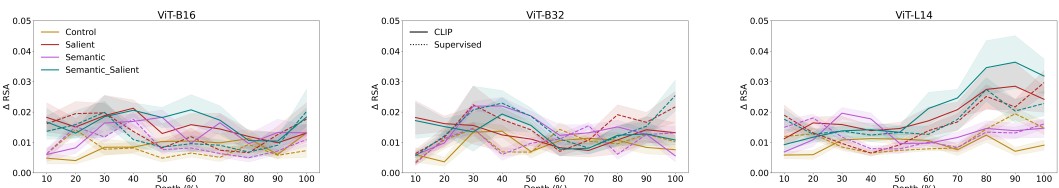

Figure 10: ΔRSA values for all tested ViTs.

### A.4 EFFECTS OF THE DISTRACTORS ON PERFORMANCE

To assess the impact of distractors on model performance, we calculated the top-5 accuracy for each model. Given that non-CLIP models were trained on ImageNet while our dataset images come from COCO, we mapped ImageNet labels to COCO labels using the method outlined at `https://github.com/howardyclo/ImageNet2COCO`.[1] For CLIP networks, we used the default prompt "A photo of a {label}" to gauge accuracy across all 1000 ImageNet labels. Since COCO images often contain multiple object classes, an image was considered correctly labeled if *any* of the COCO classes matched the network's predictions.

---

[1]We manually corrected some mismatches resulting from the hypernym-based label mapping.

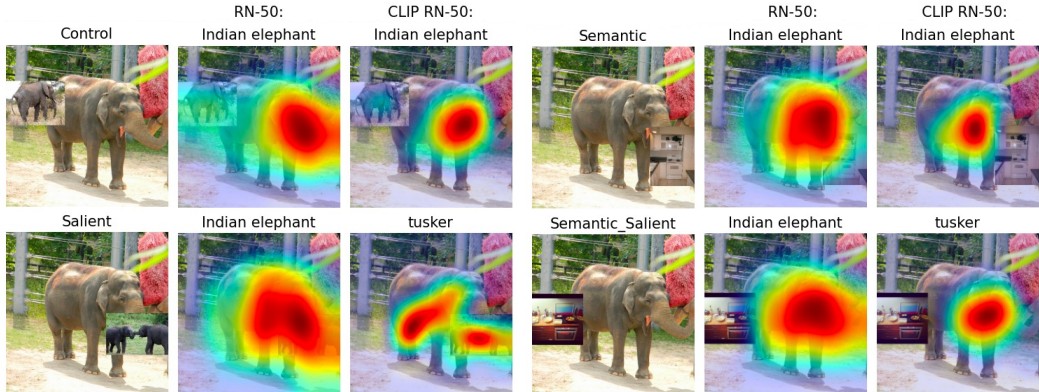

Figure 11: Grad-Cam experiment. Networks attend to the salient distractor (bottom left) more than to the control distractor (top left), even though the semantic information in both is very similar (both display elephants). Thus, the salient distractor distracts the networks when classifying the image. Nevertheless, the networks still classify the image correctly (Top 5) since semantically the target and distractor are also similar.

Table 1: Categorizing the Target Image. Top 5 Accuracy (%).

| Model | Baseline Accuracy | Distractor Condition | | | |
|---|---|---|---|---|---|
| | | Control | Salient | Semantic | Semantic & Salient |
| ResNet-50 | 41.9 | 0.7 | 0.3 | ↓ 2.6 | ↓ 1.1 |
| CLIP ResNet-50 | 41.8 | 1.1 | 0.4 | ↓ 1.5 | ↓ 3.4 |
| ViT-B16 | 42.5 | ↓ 0.3 | ↓ 0.2 | ↓ 1.9 | ↓ 1.7 |
| CLIP ViT-B16 | 43.7 | ↓ 1.4 | ↓ 0.5 | ↓ 5.7 | ↓ 5.7 |

## A.5 TRAINING DATASET SIZE: CONTROL ANALYSES

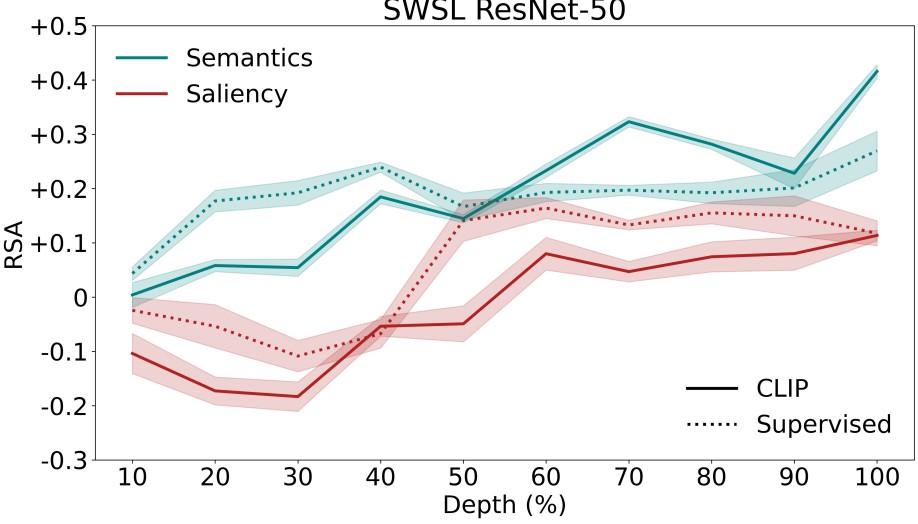

Figure 12: Control alignment analysis for SWSL ResNet-50 trained with 940M images and CLIP ResNet-50 (around 400M images). We see that the increase in dataset size does not replicate the full extent of saliency suppression as a result of CLIP.

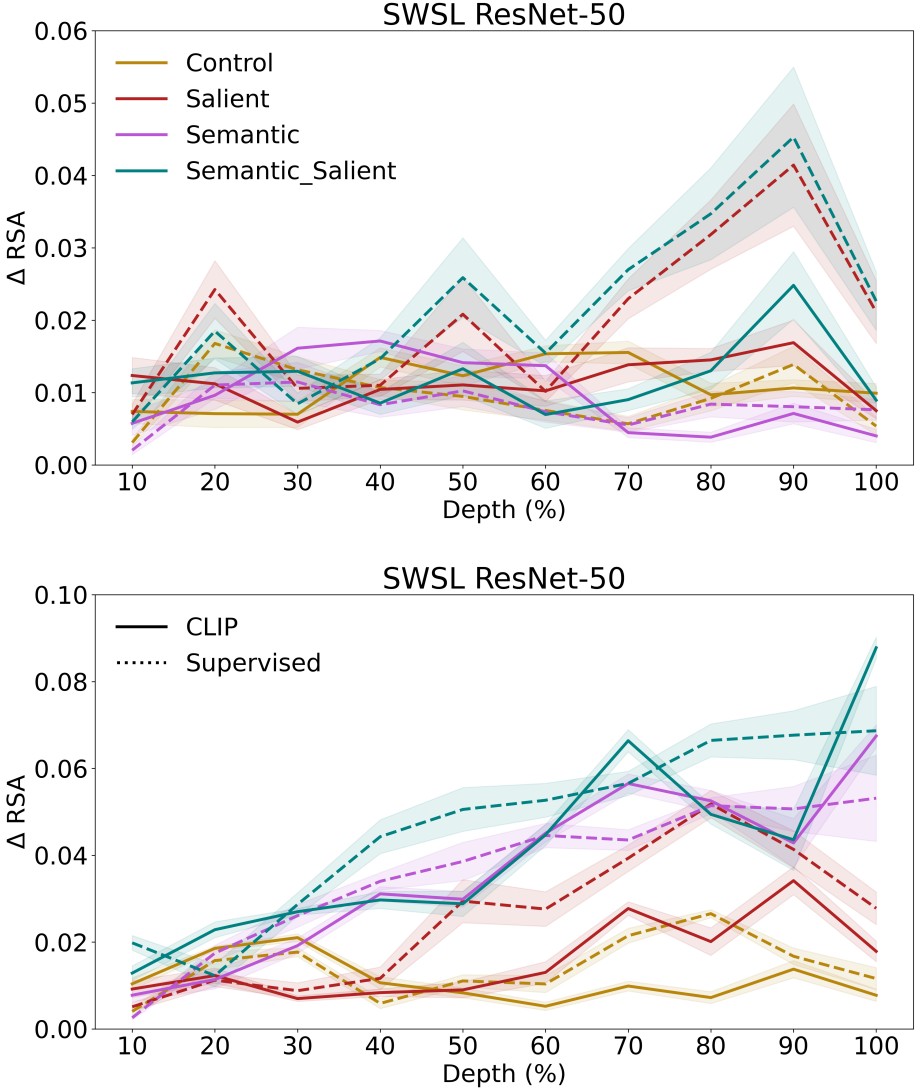

Figure 13: Control distractor analysis for SWSL ResNet-50 trained with 940M images and CLIP ResNet-50 (around 400M images). We see that the increase in dataset size does not replicate the full extent of saliency suppression as a result of CLIP (top). Similarly to the ResNet-50 trained with ImageNet (1.5M images) it is disturbed by salient distractors (see Figure 4). SWSL ResNet-50 seems to react less strongly to semantic distractors than the ImageNet-trained one (bottom).

