# OpenReview forum: "Saliency Suppressed, Semantics Surfaced: Visual Transformations in Neural Networks and the Brain"
_ICLR.cc/2024/Workshop/Re-Align — ICLR 2024 Workshop Re-Align Poster_

### Official Review · Reviewer_qXdk · 2024-02-22
**Comparison of visual transformations across networks and humans**

**Rating:** 3
**Fit:** 3
**Confidence:** 2

**Workshop Review:**

# Summary
The paper uses representational similarity analysis to understand how visual neural networks (ResNet, ViT) encode information at low (visual saliency) and high (semantic similarity) levels of abstraction. They develop a custom image dataset where they systematically manipulate salient and semantic information, providing insights into architectural and training objective differences in how neural networks perform visual transformations.  They compare visual transformations in neural networks with those in the human visual cortex, identifying areas of alignment and divergence between neural networks and the human brain, one main divergence being saliency suppression in early layers of neural networks.

# Strong points
-	Easy to follow, everything is well described and motivated
-	Interesting and novel results, comparison to humans can give more insight about how NN act similarly/differently
-	Use of newly constructed dataset that is well motivated and defined
# Weak points
-	Metrics are hard to understand if encountered for the first time
-	Colors in plots can be hard to distinguish, hard to separate overlapping lines
-	Unclear how the brain responses have been computed and handled
# Clarity
Well structured and easy to follow. Experiments clearly defined.
# Correctness
Experiment design is well adapted to the research questions. The dataset created is well described and motivated, fitting the research question.
# Novelty
Novel angle on representations alignment by looking at the influence of saliency and semantic on representations. Introduction of a new dataset to test effects of saliency and semantic information. Novel findings about saliency suppression in early layers of networks.
# Interest to community
This work is very well suited and of interest for machine learning researchers as well as the cognitive scientists and neuroscientists, as it not only looks at representations in networks but also compares it closely with brain representations and provides interesting findings of how NN and humans process visual information similar/different. This paper has potential to spark interesting discussions and potentially collaborations between neuroscientists and ML researchers to deepen the understanding of early visual processes.
# Recommendation
I recommend to accept this paper, as it is very well suited for the workshop and of interest for several groups. It shows novel results and could initiate new research collaborations.
# Recommendations for improvement
-	I would recommend putting the mathematical definition of the used scores into the main paper for easier understanding
-	Add a more detailed description of how the brain responses were handled
-	Use a colorblind friendly color scheme for plots

**Reason For Not Giving Higher Score:**

N/A

**Reason For Not Giving Lower Score:**

Interesting work with potential for fruitful discussions in the community.

**Reviewer Domain:**

machine learning

---

### Official Review · Reviewer_nxR2 · 2024-02-24
**In-depth study about encoding (and sensitivity to) different levels of visual signal**

**Rating:** 2
**Fit:** 3
**Confidence:** 2

**Workshop Review:**

Clarity and correctness: This paper is written and structured very clearly. My main concerns are related to experimental design:
1. (one writing comment) Separating Section 1 into Introduction and Related Work would make it easier to localize the motivation and contributions of this paper. Then, background about CLIP/RSA/etc can be decoupled from the storyline of the authors' saliency/semantics work.
2. Creating the saliency distractions by overlaying a smaller image onto a larger one does not seem like an obvious choice. If the definition of visual saliency is related to color, orientation, and intensity, why not do something more straightforward like photometric distortions (e.g. color distortion, blurring, NN artifacts as done in LPIPS [1])? This overlay method seems more suited to a dataset purely for semantic distortions.
3. Why is the filtering metric for saliency distortions cosine distance in color space? Perhaps MSE or metrics that measure distribution shift between two color distributions for example would be more appropriate. I may be misunderstanding this design choice though, so some justification in the paper for cosine distance here would also be helpful.
4. Figure 6: This finding with the NSD dataset is very compelling, but would be even stronger with some references to the same conclusions in neuroscience work. It feels a bit like a standalone experiment in the paper right now.

Novelty and interest: Studying how low and high level information is encoded in neural networks is not new, but the dataset and study across architectures (and architecture sizes) may be of great interest to some communities in ML/neuroscience.

References:

[1] Zhang, Richard et al. “The Unreasonable Effectiveness of Deep Features as a Perceptual Metric.” 2018 IEEE/CVF Conference on Computer Vision and Pattern Recognition (2018): 586-595.

**Reason For Not Giving Higher Score:**

See concerns regarding experimental design above. While they don't negate the interesting conclusions and dataset contribution of the paper (hence the final rating), some choices are not well justified in the paper.

**Reason For Not Giving Lower Score:**

This paper meets the criteria for the workshop, and will likely be of interest to ML and neuroscience researchers.

**Reviewer Domain:**

machine learning

---

### Decision · Program_Chairs · 2024-03-02

Accept (Poster)